# Addition of Radiotherapy to Immunotherapy: Effects on Outcome of Different Subgroups Using a Propensity Score Matching

**DOI:** 10.3390/cancers12092429

**Published:** 2020-08-27

**Authors:** Maike Trommer, Jaika Kinsky, Anne Adams, Martin Hellmich, Max Schlaak, Michael von Bergwelt-Baildon, Eren Celik, Johannes Rosenbrock, Janis Morgenthaler, Jan M. Herter, Philipp Linde, Cornelia Mauch, Sebastian Theurich, Simone Marnitz, Christian Baues

**Affiliations:** 1Department of Radiation Oncology and Cyberknife Center, University Hospital of Cologne, Kerpener Str. 62, 50937 Cologne, Germany; jaika.kinsky@uk-koeln.de (J.K.); eren.celik@uk-koeln.de (E.C.); johannes.rosenbrock@uk-koeln.de (J.R.); janis.morgenthaler@uk-koeln.de (J.M.); jan.herter@uk-koeln.de (J.M.H.); philipp.linde@uk-koeln.de (P.L.); simone.marnitz-schulze@uk-koeln.de (S.M.); christian.baues@uk-koeln.de (C.B.); 2Radio Immune-Oncology Consortium (RIO), University Hospital of Cologne, Kerpener Str. 62, 50937 Cologne, Germany; max.schlaak@med.uni-muenchen.de (M.S.); michael.bergwelt@med.uni-muenchen.de (M.v.B.-B.); sebastian.theurich@med.uni-muenchen.de (S.T.); 3Center for Integrated Oncology (CIO), University Hospital of Cologne, Kerpener Str. 62, 50937 Cologne, Germany; cornelia.mauch@uk-koeln.de; 4Center for Molecular Medicine Cologne (CMMC), University Hospital of Cologne, Kerpener Str. 62, 50937 Cologne, Germany; 5Institute of Medical Statistics and Computational Biology, University of Cologne, Faculty of Medicine and University Hospital of Cologne, Kerpener Str. 62, 50937 Cologne, Germany; anne.adams@uni-koeln.de (A.A.); martin.hellmich@uni-koeln.de (M.H.); 6Department of Dermatology and Allergology, LMU University Hospital, Ludwig-Maximilians University (LMU), Munich, Frauenlobstr. 9-11, 80377 Munich, Germany; 7Department III of Internal Medicine, LMU University Hospital, Ludwig-Maximilians University (LMU), Munich, Marchioninistr. 15, 81377 Munich, Germany; 8Department of Dermatology and Allergology, University Hospital of Cologne, Kerpener Str. 62, 50937 Cologne, Germany; 9Cancer & Immunometabolism Research Group, Gene Center LMU, Ludwig-Maximilians University, Munich, Feodor-Lynen-Str. 25, 81377 Munich, Germany

**Keywords:** PD-1/PD-L1, immune checkpoint inhibition, radiotherapy, radioimmunotherapy, combination treatment, subgroups, propensity score matching

## Abstract

Immune checkpoint inhibition (ICI) has been established as successful modality in cancer treatment. Combination concepts are used to optimize treatment outcome, but may also induce higher toxicity rates than monotherapy. Several rationales support the combination of radiotherapy (RT) with ICI as radioimmunotherapy (RIT), but it is still unknown in which clinical situation RIT would be most beneficial. Therefore, we have conducted a retrospective matched-pair analysis of 201 patients with advanced-stage cancers and formed two groups treated with programmed cell death protein 1 (PD-1) inhibitors only (PD1i) or in combination with local RT (RIT) at our center between 2013 and 2017. We collected baseline characteristics, programmed death ligand 1 (PD-L1) status, mutational status, PD-1 inhibitor and RT treatment details, and side effects according to the Common Terminology Criteria for Adverse Events (CTCAE) v.5.0. Patients received pembrolizumab (*n* = 93) or nivolumab (*n* = 108), 153 with additional RT. For overall survival (OS) and progression-free survival (PFS), there was no significant difference between both groups. After propensity score matching (PSM), we analyzed 96 patients, 67 with additional and 29 without RT. We matched for different covariates that could have a possible influence on the treatment outcome. The RIT group displayed a trend towards a longer OS until the PD1i group reached a survival plateau. PD-L1-positive patients, smokers, patients with a BMI ≤ 25, and patients without malignant melanoma showed a longer OS when treated with RIT. Our data show that some subgroups may benefit more from RIT than others. Suitable biomarkers as well as the optimal timing and dosage must be established in order to achieve the best effect on cancer treatment outcome.

## 1. Introduction

Immunotherapy (IT) with immune checkpoint inhibitors (ICI) has changed oncologic treatment strategies dramatically even in hard-to-treat advanced cancers like malignant melanoma (MM), non-small cell lung cancer (NSCLC), squamous cell carcinoma of the head and neck (SCC), or metastatic urothelial carcinoma [1,2,3,4]. Highest response rates including long-term remissions are currently achieved by ICI combinations such as the anticytotoxic T-lymphocyte-associated protein 4 (CTLA-4) antibody ipilimumab with a programmed cell death protein 1 (PD-1) inhibitor [5,6]. However, due to significantly increased immune-related adverse events (IrAE), alternative approaches that enhance antitumor responses but spare IrAE are urgently needed and topic of intensive research efforts. One promising approach that enhances antitumor immune response while sparing toxicity might be the combination of ICI with local radiotherapy (RT) [7,8,9]. RT-induced systemic immunological effects have been discussed for a long time. Local application of RT to the tumor induces effects that go beyond the killing of tumor cells at that exact site and send distinct signals to the host’s immune system. Several mechanisms have been described how RT influences the tumor microenvironment and how it may increase antitumor immune responses [8,10,11,12,13]. A subsequent discussion about synergistic effects was encouraged. Clinical reports about abscopal effects, the shrinkage of nonirradiated lesions as sign of a systemic-mediated effect of RT, have been published since decades [14]. However, due to their rare occurrence in the pre-ICI area, abscopal effects might have been underestimated in clinical routine so far. Recent results from both, preclinical and clinical studies involving ICI therapy have shown that abscopal effects are seen in this particular immune stimulating setting and could be further exploited therapeutically [15,16,17,18,19].

RT can not only increase the expression of programmed death ligand 1 (PD-L1) on tumor cells but also inhibit T-cell activation along the PD-1/PD-L1 axis, so that specific anti-PD-1 antibodies can then interact at this point [20,21,22]. DAMPS (danger-associated molecular patterns) are also released by RT, which leads to immune activation via antigen presenting cells [23]. This effect can be significantly enhanced by combination with checkpoint inhibitors [8,20,21,22].

Clinical results of RIT support these preclinical rationales: In the so-called Pacific Trial, PD-L1 inhibitor consolidation after RT showed a significantly prolonged progression-free survival (PFS) and a better overall survival (OS) compared to the placebo group [24,25].

However, markers that would allow the identification of patients who would benefit most from combined RIT are still under investigation. Therefore, we conducted this retrospective study using a propensity score matched-pair analysis from a patient cohort undergoing PD-1 inhibition with or without additional RT at the University Hospital of Cologne. Besides established risk factors associated with shortened survival time, such as the presence of brain metastases and a low performance status, we have investigated other covariates that potentially affect ICI treatment outcome. These were cancer type, smoking status, and multiple metastases, all of which were associated with high mutational load [26,27,28]. Furthermore, we included previous treatment with ipilimumab and obesity, both associated with inflammation [29,30].

## 2. Results

### 2.1. Patients and Treatment Characteristics

Our database consisted of 209 patients who were treated with anti-PD-1-antibodies from 2013 to 2017 at the University Hospital of Cologne. Overall, 201 cases were sufficiently documented and provided sufficient data quality for analysis. Baseline characteristics for the whole cohort are shown in Table 1.

In a propensity score matching (PSM), we adjusted for variables that had a potential impact on OS to minimize differences in baseline characteristics. Patients without a matching partner were excluded. After PSM, we had a total of 96 patients to analyze. In the following paragraph, the results after matching are shown in brackets.

We analyzed 41.3% (38.5%) female and 58.7% (61.5%) male patients with a mean age of 62.4 ± 13.2 (65.5 ± 12.8) years. 48% (51.6%) were diagnosed with malignant melanoma, 34% (26.3%) with non-small cell lung cancer, 7% (4.2%) with renal cell carcinoma (RCC), and 11% (18%) with other tumor types (small cell lung cancer (SCLC), urothelial carcinoma, esophageal cancer, head and neck cancer, Hodgkin’s lymphoma and other). Most patients (98.4%) were diagnosed with advanced disease in UICC/AJCC stage III and IV and had multiple metastases but a favorable Eastern Cooperative Oncology Group (ECOG) performance status between 0 and 1. PD-L1 status, if assessed, was positive in 48 patients (62.3%) and negative in 29 patients (37.7%). After matching, we found 23 patients (59%) PD-L1 positive and 16 patients (41%) PD-L1 negative. More patients were ipilimumab-naïve and received no steroids during treatment. More than half of the patient cohort had a BMI > 25, and patients tended to be nonsmokers.

After splitting the patients into the two different treatment groups, RIT and PD1i, statistically significant differences were observed concerning age (*p* = 0.038), brain metastases at baseline (*p* < 0.001), and prior therapy with ipilimumab (*p* = 0.033) with more brain metastases, preliminary ipilimumab therapy, and a younger age in the RIT group. Baseline characteristics, including ICI and RT details for the treatment groups before and after PSM are shown in Table 2. 

To show the balance comparison, we displayed z-differences before and after PSM in Figure 1. See the balancing of baseline characteristics before and after PS matching in Appendix A
Table A1.

### 2.2. Immunotherapy and Radiotherapy

In particular, 46.3% (49%) of the analyzable patients received pembrolizumab, 53.7% (51.0%) nivolumab, the median ICI duration was 17.1 (6.4–44.0) (17.1 (6.1–38.1)) weeks, and patients received at least 3 (3) cycles. Precisely, 153 (67) patients were found to receive additional local RT and were assigned to the RIT group. In case patients received more than one schedule of PD-1 inhibitors in their medical history, we referred to the ICI treatment given closest to the RT performed.

Additionally, 84 (32) patients received more than one course of RT, so we defined the RT that was applied closest to the given anti-PD-1 therapy as “RT of interest” and reported on this RT in detail. Conventionally fractionated RT (CFX) was applied in 34.1% (44.8%) of the cases, *n* = 46 (26). Hypofractionated treatment (HFX) was overrepresented with 40.7% (41.4%), *n* = 55 (24) and SBRT and SRS (stereotactic body RT and stereotactic radiosurgery) was applied in 25.2% (13.8%) of the cases, *n* = 34 (8). Most patients were irradiated for bone or soft tissue metastases, cerebral metastases, or lymph node metastases. Details for ICI and RT treatment are presented in Table 2.

### 2.3. Outcome Evaluation

For outcome evaluation, we report medians if possible. If the survival did not drop to 50% or below, we reported the means. If the numbers of patients per group in the tables, figures, and in the text differ, it is due to the missing events.

#### 2.3.1. Unmatched Cohort

We obtained a median follow-up period of 19.3 months and a median overall survival time of 14.1 months for the entire cohort. The PD1i group reached a median follow-up time of 18.6 months, whereas for the RIT group, it was 20.6 months. By the time of data cutoff, 23 patients (50.0%) in the PD1i group and 84 (56.8%) in the RIT group had died. 

The median OS was 20.4 months (95% CI, 4.4–36.5) for the PD1i and 14.1 months (95% CI, 8.9–19.4) for the RIT group (HR 1.05 (95% CI, 0.66–1.67); *p* = 0.828) (see Figure 2A). The 12-month overall survival rate was 54.1% in the PD1i group and 52.8% in the RIT group. By the time of data cutoff, the median PFS duration was 5.9 months (95% CI, 1.6–10.3) in the PD1i group and 3.8 months (95% CI, 1.8–5.8) in the RIT group (HR 1.09 (95% CI, 0.74–1.61); *p* = 0.671) (see Figure 2B).

In the log-rank test, we observed no statistically significant differences for OS or PFS between the two groups or—when considering the different covariates that had an impact on overall survival across the entire cohort—in some cases, the group sizes were too small to make a valid statement. PD-L1-positive patients in the RIT group tended to have a better overall survival with a median overall survival time of 11.3 vs. 4.8 months (*p* = 0.050) (see Figure 2C). Regarding three cut-offs for PD-L1 expression (<1%; 1 < 50%; ≥50%), patients in the RIT group tended to show a better OS with a PD-L1 expression of ≥1%–<50% (*p* = 0.084) (see Figure A1 in Appendix A).

In the entire unmatched cohort, all other curves considering the different subgroups run almost parallelly in OS and PFS functions when comparing the PD1i and RIT group. 

There is a trend towards a better OS for patients being treated simultaneously regarding conventional fractionated (CFX) or stereotactic (SRS) RT than for patients being treated sequentially (see Figure A2 in Appendix A).

Kaplan–Meier curves of the entire cohort are shown in Figure 2.

#### 2.3.2. Matched Cohort

After PS matching, the RIT group mainly lost younger patients, patients who had brain metastases at baseline or were previously treated with ipilimumab. The median OS of the entire matched cohort was 12.4 months. By the time of the data cutoff, 16 patients (57.1%) in the PD1i group and 39 (60.0%) in the RIT group had died. 

The PD1i group reached a median OS of 8.5 months, whereas the RIT group, reached a median of 14.1 months (HR 0.89 (95% CI, 0.50–1.59); *p* = 0.692). The 12-month overall survival rate was 41.6% in the PD1i group, as compared to 54.9% in the RIT group. The 24-month overall survival rate was 41.6% in the PD1i group and 31.3% in the RIT group. The PD1i group reaches a survival plateau at 10 months after the start of PD-1 inhibitor treatment. The RIT curve runs above the PD1i curve, crossing its plateau at 17.5 months, see Figure 3A. The median PFS duration was 3.7 months (95% CI, 1.2–6.2) in the PD1i group and 3.8 months (95% CI, 1.7–5.9) in the RIT group ((HR 0.90 (95% CI, 0.54–1.51); *p* = 0.692) (data not shown). 

Considering the different covariates, patients with a positive PD-L1 status (*n* = 23) showed a median OS of 16.6 months in the RIT group and 4.8 months in the PD1i group, *p* = 0.055 (see Figure 3B). PD-L1-negative patients (*n* = 16) had a median OS time of 14.1 (RIT) and 8.5 (PD1i) months, *p* = 0.948 (data not shown).

Smokers (*n* = 24) showed a significantly better OS when being in the RIT group (*p* = 0.029) (see Figure 3C), whereas there was no difference for nonsmokers (see Figure 3D).

Patients with a BMI ≤ 25 (*n* = 33) had a statistically significant better OS in the RIT group with a mean OS of 21.9 months and 6.4 months in the PD1i group, *p* < 0.001, see Figure 3E. The group of patients with a BMI > 25 (*n* = 63) had a mean OS of 13.9 months in the RIT and 25 months in the PD1i group (median: 11.7 months) with a plateau in the PD1i group at 10 months, *p* = 0.052 (see Figure 3F).

Regarding tumor entities, malignant melanoma patients (*n* = 48) in the PD1i group reached a survival plateau at 8.5 months with a mean OS time of 31.6 months vs. 16.9 months when being irradiated, *p* = 0.027 (see Figure 3G). Patients without malignant melanoma (*n* = 45) had a statistically significant better OS in the RIT group (median OS of 14.1 (RIT) and 3.9 (PD1i) months, *p* = 0.005) (see Figure 3H).

Ipilimumab-naïve patients reached a mean OS time of 11.7 (RIT) and 9.9 (PD1i) (data not shown).

Patients without brain metastases reached a median OS of 14.1 (RIT) and 9.2 (PD1i) months, *p* = 0.776 (data not shown). The patient group with brain metastases consisted of only 5 patients after PSM.

Median survival time for patients with multiple metastases (*n* = 65) was 16.6 months (RIT) and 8.5 months (PD1i), *p* = 0.653 (data not shown). 

Regarding BRAF and NRAS mutational status, there were not enough patients to make a valid statement (*n* = 15 BRAF positive, *n* = 15 NRAS positive).

Kaplan–Meier survival curves of the matched cohort are shown in Figure 3.

### 2.4. Safety

Regarding all reported events, more side effects were reported when adding RT to the ICI treatment (9% vs. 19%). 

Expected immunotherapy-related side effects such as pulmonary, thyroid, and liver toxicities did not differ significantly in between both groups. We did not observe differences in severe toxicities, grade 3 and 4 events were evenly distributed in both treatment groups (3.3% and 3.2%). We observed no treatment-related deaths. Treatment-related toxicities are summarized in Appendix A, Table A2 and Table A3.

## 3. Discussion

In this study, we aimed to identify clinical factors that might predict improved response to additional RT in a cohort of 201 patients with mixed primary diagnosis undergoing ICI treatment. After a matched-pair analysis, patients with a combined RIT tended to show a better 12-month OS rate than patients treated with ICI alone. We were able to identify certain subgroups that benefit more than others from the addition of RT. These include a positive smoking status, a BMI less than or equal to 25, and a positive PD-L1 status.

Using a propensity score matching, we compared two groups (PD1i and RIT) based on different covariates that had a potential impact on outcome (age, ECOG, brain metastases at baseline, smoking status, BMI, previous therapy with ipilimumab, and treatment with betablockers). Due to this statistical tool, we were able to improve the quality of this retrospective study.

In the OS and PFS analyses of the entire cohort, we could not identify a clear clinical benefit when comparing patients with PD-1 inhibitors only with patients with additional RT. Before the PSM was performed, the RIT group contained a lot more patients with brain metastases at baseline (34% vs. 4.2%) and less ipilimumab-naïve patients (22.5% vs. 8.5%). Since patients with brain metastases frequently receive RT as part of their treatment, a higher number of them was to be expected in the RIT group. With respect to the higher proportion of patients with previous ipilimumab therapy in the RIT group, it may be assumed that in this group, more patients with CTLA-4 therapy had a relapse and had to switch to another therapy. Additionally, the RIT group contained patients with poorer ECOG scores.

Therefore, the trend for a better OS in the RIT group with a median of 11.3 vs. 4.8 months (*p* = 0.050) for PD-L1-positive patients (Figure 2C) is remarkable. 

After PSM, the OS for the RIT group with a median of 14.1 months is superior to the OS for the PD1i group with a median of 8.5 months (Figure 3A). Regarding 12-months OS rate, the RIT group shows a clear trend for a better OS. Nevertheless, OS curves are crossing at approximately 17.5 months and the PD1i group shows a plateau. Based on clinical data, there are several options for a therapeutic response to ICI: patients who continue responding (responders), those who never respond (innate resistance), and others who respond but subsequently developed disease progression (acquired resistance) [31,32,33]. Nonresponders or never-responders may benefit from combination therapies such as complementary RT to ICI therapy. This aspect is further discussed below. 

We were able to show the impact of various parameters on the efficacy of RIT. Some of them show significant advantages.

The benefit for smokers, e.g., Figure 3C, is very interesting, as there is data showing that treatment of PD-1 inhibitors was more effective in smokers than nonsmokers [34,35]. We were able to show that adding RT to PD-1 inhibition might even enhance this, here exceptional, advantage for smokers. Overall, there is little evidence on the relationship between smoking, the effectiveness of PD-1/PD-L1 inhibitors, and immune cells. The consensus in literature suggests that a higher PD-L1 tumor proportion score (≥50%) and mutational burden is correlated with a positive smoking history, and that smokers show better overall response rates of ICI than nonsmokers [28,36]. The lack of data from analyses of smoker status in many trials does not allow definite conclusions to be drawn at this stage, but available data suggest that future studies should investigate the influence of smoking status on the response to PD-1 treatment.

Another interesting aspect is the finding that patients with a BMI greater than 25 did not benefit from combined RIT (Figure 3F), whereas patients with a BMI less than or equal to 25 showed significantly longer survival when being irradiated (Figure 3E). BMI values were balanced, with no statistical differences in underweight, normal weight, or obese patients between both treatment groups (Table 2 and Table A1). This finding can, therefore, not be attributed to an increased number of obese or underweight patients in any of the groups. Mean OS did not differ between patients with BMI ≤ 25 (17.2 months) or > 25 (17.0 months) in the entire patient cohort. The question arises why the outcome with RIT is superior in patients with BMI ≤ 25 (mean OS 21.9 months and only 6.4 with anti-PD-1 monotherapy) and why the outcome with RIT appears worse in patients with BMI > 25 (mean OS 13.9 months and 25 with anti-PD-1 monotherapy). Whether and how the interaction of the potential chronic inflammation of these patients and a benefit of immunotherapy could be negatively influenced by RT is unclear. There is clinical evidence that cancer patients with obesity might benefit from anti-PD-1/PD-L1 immune checkpoint inhibitors [37,38]. The authors conclude that obesity could be regarded as a tumorigenic immune deficiency that could be overcome by checkpoint inhibition. They suggest BMI as a valuable prognostic tool in clinical practice and as a stratification factor in prospective clinical trials. In obese patients, PD-1 inhibition alone might be more effective than in combination with RT. 

Besides patients with a high BMI, neither did patients with malignant melanoma benefit from RIT (Figure 3G). There is recent retrospective data stating that previous treatment with RT may not be beneficial in patients with metastatic malignant melanoma undergoing ICI treatment [39]. There are several reports on combination therapies in patients with malignant melanoma treated with ipilimumab targeting the CTLA-4 checkpoint, as this was already approved in 2011 for the treatment of metastatic melanoma [40]. Amongst others, Theurich et al. showed in 2016 that the addition of a local peripheral treatment (in 89% of the cases applied as RT) to ipilimumab in advanced malignant melanoma patients prolongs overall survival significantly [41,42,43]. Statistically significant differences regarding the timing of the application of RT were not detected [41]. This patient group must certainly be considered in further studies on combination treatments. It must be assumed that the patients in this study who received RIT had a higher disease burden than those who received immunotherapy alone. Therefore, a statistical weakness at this point is not unlikely. It is all the more surprising as various subgroups performed better with RIT than with ICI alone. Their prognosis may have improved considerably with the combined therapy.

Our toxicity data analysis is in line with previous reports showing that the combination of RT and PD-1 inhibition is safe and feasible [24,44]. The RIT group shows more side effects, but grade 3 and 4 toxicities are balanced in both groups. The known immunotherapy-related side effects are not statistically significantly increased in patients who were treated in combination. We observe statistical differences with more reported toxicities in the RIT group for skin toxicities, fatigue, vertigo, and gastrointestinal problems including nausea, all of which are known side effects of RT. 

In 2014, checkpoint inhibitors targeting the PD-1 receptor were approved for the treatment of melanoma and their indications were extended to a broad spectrum of cancers such as non-small cell lung cancer, renal cell carcinoma, bladder cancer, head and neck squamous cell carcinoma, MSI-high colorectal carcinoma, Merkel cell carcinoma, and Hodgkin lymphoma [45,46]. The interaction of PD-1 and its ligand PD-L1, which may be expressed on tumor cells and antigen presenting cells (APCs), leads to a suppression of T-cell activation and thus provides an immune escape for cancer cells [47]. RT can have the undesirable effect of inducing the expression of PD-L1 on these cells [20]. Here, treatment with anti-PD-L1/anti-PD-1 antibodies proves beneficial and eventually leads to the induction of antitumor immunity [48]. In our patient cohort, PD-L1-positive patients benefited from the addition of RT. PD-L1-negative patients also had a better median OS time with RIT, but especially in this covariate too many values were missing to provide sufficient statistical power. 

As mentioned above, there is still a considerable number of patients who do not respond to mono-ICI treatment at all, but only achieve a partial response or relapse. In patients with innate resistance, the CD8+ T cells could be either unable to recognize and localize the tumor or become ineffective. The mechanisms of acquired resistance are complex and diverse and may include loss of T-cell function, interruption of antigen presentation, and resistance to interferon generated by T-cells [49]. If there is a weak endogenous immune response, an upregulation of PD-L1 on tumor cells, is unlikely. In this case, anti-PD-L1/anti-PD-1 therapies are ineffective [50]. 

A hypothesis to overcome this resistance to ICI treatment could be the initiation of a treatment that provides new targets for the immune system to attack. Local RT is able to provide those targets by mechanisms like antigen release, cytokine production, complement activation, increasing the expression of the major histocompatibility complex (MHC) class I, activating dendritic cells, enhancing the presentation of tumor antigens, and the migration of immune cells into the tumor microenvironment. This leads to an increase of tumor-infiltrating lymphocyte density with a broader T-cell receptor repertoire, improved effector T cell activity, and modulation of TReg cells [12,22,51,52]. 

The effects of RT on the immune system have been much better described and understood in recent years, and yet it is still unclear how the two modalities can be combined in the best possible way. The main questions here are the individual and total dose of RT and the optimal time sequence of both therapies. RT-induced cancer cell death releases signals, which lead to the attraction of immune cells like macrophages [53]. Those not only play a major role in the tumor microenvironment as partners for cancer cell migration and metastasis due to their function as inflammatory stromal component [54] but also have the potential to act as APC to induce an adaptive T-cell immune reaction [55]. Conventional fractionation induces inflammation and macrophage migration. However, Pinto et al. demonstrated that macrophages irradiated with 2 Gy per fraction could also promote cancer cell invasion and cancer cell-induced angiogenesis [56]. Low doses of 0.5–1.0 Gy per fraction are usually used to treat inflammatory diseases [57], and in this dose-range, immunosuppressive effects on macrophages are particularly evident [58]. A similarly complex effect seems to occur at very high single doses (>15 Gy). On the one hand, highly antigen-presenting dendritic cells are activated [59], on the other hand, immunosuppressive repair enzymes are formed [60]. The question is which mechanisms induced by different dosage and fractionation schemes in the tumor microenvironment are most appropriate for a combination of ICI to be the best complement. 

Regarding the right timing of ICI and RT application, Dovedi et al. were able to show in a preclinical study that the time sequence of the combined therapy has a decisive influence on the effect of the systemic response. Thus, the group was able to demonstrate that in the best case, immediate simultaneous treatment in mice with flank tumor irradiation induced the strongest systemic effect in terms of an abscopal response [22].

Due to its retrospective character and heterogeneous patient collective, our analysis has statistical weaknesses. By propensity score matching, we could improve the data quality, resulting in a smaller sample size. To verify the effects of combined RIT and establish missing predictive surrogate parameters for treatment response and benefit, large randomized trials are needed. 

## 4. Materials and Methods 

### 4.1. Patients and Treatment

We identified 209 patients treated with the PD-1 inhibitors pembrolizumab or nivolumab at the University Hospital of Cologne between May 2013 and December 2017 (*n* = 209). From them, 201 patients were sufficiently documented and could be enrolled in this retrospective study. This research has been approved by the Ethics Committee of the University of Cologne, Faculty of Medicine (reference: 19-1160). We analyzed two groups of patients. Group one had received PD-1 inhibitors (PD1i) only. The other group had received PD-1 inhibitors and additional RT during the course of their disease (RIT), regardless of the timing of both treatments.

We included any irradiation concept with respect to fractionation scheme and irradiation dose. Possible fractionation schemes were the conventional fractionated radiation therapy (CFX), which is between 1.8 and 2 Gy single dose with 5 fractions per week, the hypofractionated radiation therapy (HFX) with higher irradiation doses between 2.5 and 4 Gy single dose and less fractions; the stereotactic body RT (SBRT); and the stereotactic radiosurgery (SRS) with ablative single doses between 9 and 20 Gy and one or very few fractions as very precise irradiation of small tumor volumes. Nivolumab, 3 mg/kg, was given intravenously every 2 weeks, whereas pembrolizumab, 2 mg/kg, was given every 3 weeks.

By collecting patient data, we defined a baseline status at the start of the ICI treatment. Data recorded included baseline demographics, ECOG performance status, cancer type and stage, brain metastases at baseline, PD-L1 status, prior therapy with anti-CTLA-4 antibodies (ipilimumab), betablocker and steroid therapy, smoking status, body mass index (BMI), multiple metastases (≥2 metastatic sites), fractionation scheme and localization of RT, details of PD-1 inhibitor treatment (type and applied cycles), and side effects of the RT and/or anti PD-1 treatment according to CTCAE v5.0. 

To calculate follow-up periods, we used the reverse Kaplan–Meier estimator by Schemper and Smith [61].

### 4.2. Outcome Evaluation

OS was defined as time from start of the PD-1 inhibitor treatment to death of any cause. PFS was defined as the time from start of the PD-1 inhibitor treatment to disease progression or disease-related death. We set the PFS-event date as the radiological image with progression taken before change of treatment or disease-related death. Nonevent cases were censored in PFS and OS analyses. We defined those cases as patients alive who did not show radiological progression at the last visit. Radiological outcome was measured using radiological images according to RECIST (Response Evaluation Criteria in Solid Tumors) version 1.1 and iRECIST [62].

### 4.3. Statistical Analysis

A propensity score matching (PSM) was performed to match both treatment groups according to patient characteristics. For PSM, all covariates that could have an influence on outcome were included (age, ECOG, brain metastases, smoking status, BMI, previous therapy with ipilimumab, and therapy with betablockers). We measured covariate balance by calculating the z-difference [63]. Statistical analyses were performed separately for the entire patient cohort and for the PSM cohort.

All statistical analyses were performed using SPSS v. 24 (IBM Corp, Armonk, NY, USA) and R v. 3.4.3 (R Core Team, Vienna, Austria: R Foundation for Statistical Computing, 2017). Patient and disease characteristics, as well as treatment-related toxicities, were compared by the Kruskal–Wallis test for continuous variables and Pearson’s chi-square test for categorical variables where appropriate. PFS and OS were estimated by the Kaplan–Meier method and curve comparisons were calculated using the log-rank test. Multivariable Cox proportional hazards regression was performed to evaluate the effect of multiple covariates simultaneously on OS. In any case, *p*-values < 0.05 were considered significant and refer to two-sided tests. 

## 5. Conclusions

Several factors across different malignancies seem to positively influence the response to combined radioimmunotherapy. In our analysis, patients with a BMI ≤ 25, smokers, and PD-L1-positive patients benefit more from combined treatment than others. We could also show that combined radioimmunotherapy can be safely applied to different malignant entities and localizations. Based on promising preclinical data, establishing suitable markers and finding the optimal schedule, fractionation, and dosage for the application of radioimmunotherapy in clinical routine remains a challenge.

## Figures and Tables

**Figure 1 cancers-12-02429-f001:**
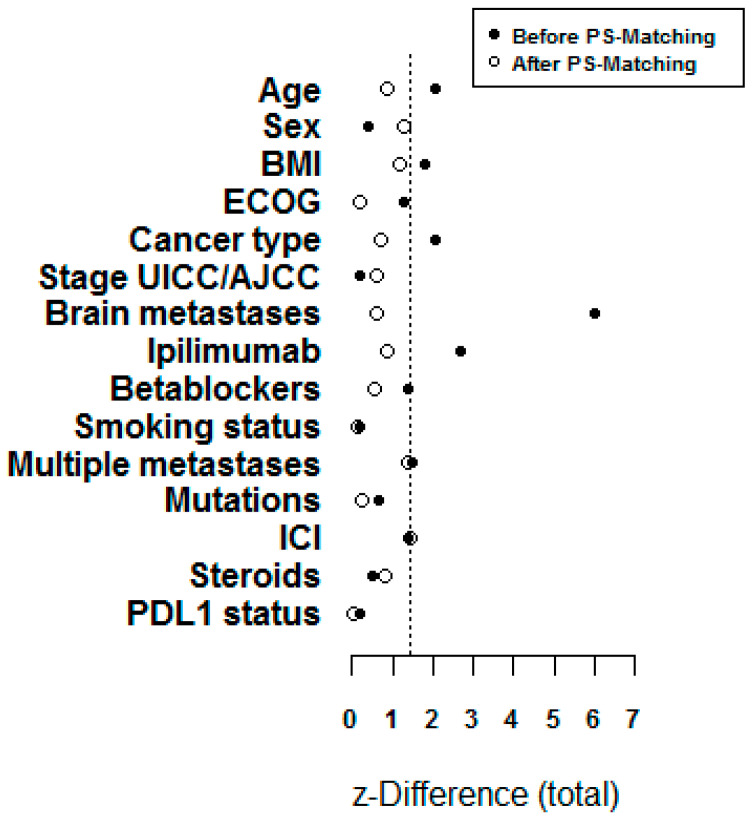
Z-differences before and after propensity score matching.

**Figure 2 cancers-12-02429-f002:**
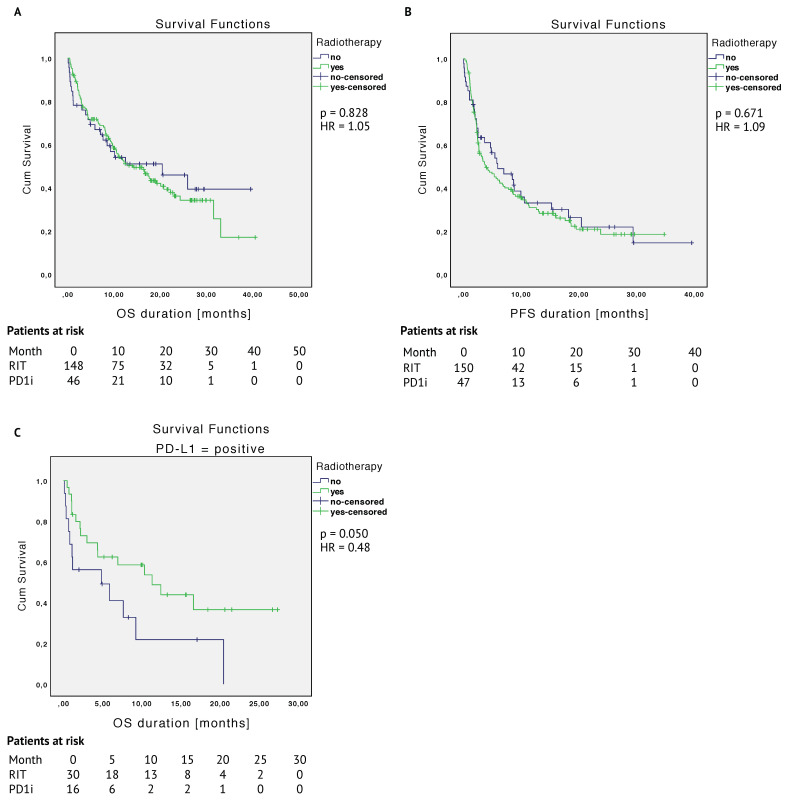
Kaplan–Meier survival curves comparing radioimmunotherapy (RIT) and immunotherapy alone (PD1i)—entire cohort. (**A**): OS = overall survival, (**B**): PFS = progression-free survival, and (**C**): OS differences in the programmed cell death 1 ligand 1 (PD-L1)-positive patient group.

**Figure 3 cancers-12-02429-f003:**
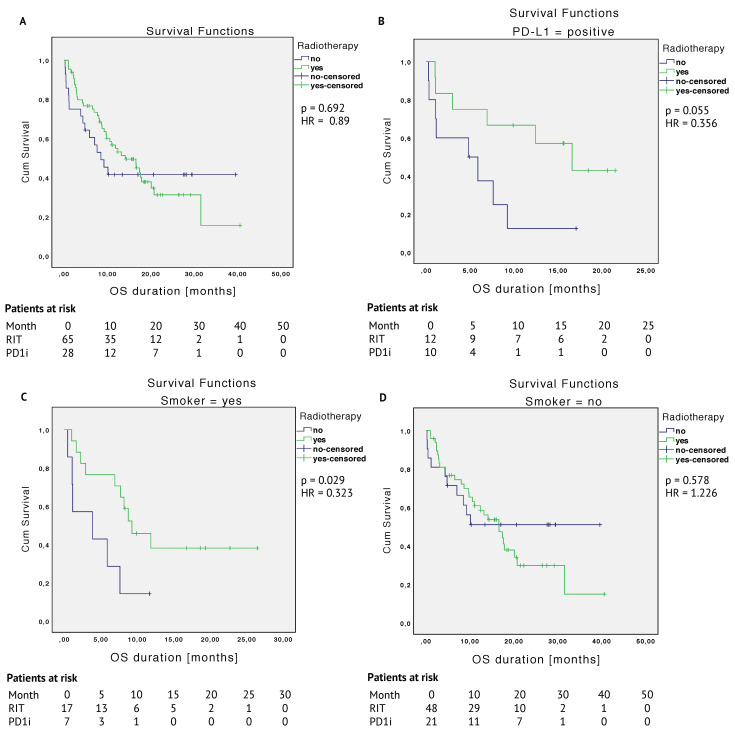
Kaplan–Meier survival curves comparing radioimmunotherapy (RIT) and immunotherapy alone (PD1i)—matched cohort. (**A**): OS = overall survival, (**B**): OS differences in the PD-L1-positive patient group, (**C**): OS differences in smokers, (**D**): OS differences in nonsmokers, (**E**): OS differences in patients with a BMI ≤ 25, (**F**): OS differences in patients with a BMI > 25, (**G**): OS differences in the patient group with malignant melanoma, and (**H**): OS differences in the patient group without malignant melanoma.

**Table 1 cancers-12-02429-t001:** Baseline characteristics.

Characteristic	All Patients *n* = 201
Age (years)	62.4 ± 13.2
Sex	
Female	83 (41.3)
Male	118 (58.7)
ECOG	
0	72 (37.3)
1	88 (45.6)
2	23 (11.9)
3	7 (3.6)
4	3 (1.6)
Cancer type	
MM	96 (48.0)
NSCLC	68 (34.0)
RCC	14 (7.0)
Other	22 (11.0)
Stage UICC/AJCC	
2	3 (1.6)
3	15 (8.0)
4	169 (90.4)
Multiple metastases (yes > 1 site)	
Yes	145 (79.2)
No	38 (20.8)
PD-L1 status	
Positive	48 (62.3)
Negative	29 (37.7)
Brain metastases	
Yes	54 (26.9)
No	147 (73.1)
Prior ipilimumab	
Yes	38 (19.2)
No	160 (80.8)
Smoker	
Yes	45 (25.3)
No	133 (74.7)
BMI	
≤25	73 (36.3)
>25	109 (54.2)

Baseline characteristics of all patients. Continuous variables are reported as mean ± SD (normal distributed), categorical variables as *n* (%). Percentages are adjusted for missing values. ECOG = Eastern Cooperative Oncology Group; MM = malignant melanoma; NSCLC = non-small cell lung cancer; RCC = renal cell carcinoma; PD-L1 = programmed death ligand 1; BMI = body mass index.

**Table 2 cancers-12-02429-t002:** Baseline characteristics for both treatment groups.

Characteristic	Unmatched		Matched			
	Radiotherapy		Radiotherapy			
	yes (RIT) *n* = 153	no (PD1i) *n* = 48	*p* Value	yes (RIT) *n* = 67	no (PD1i) *n* = 29	*p* Value
Age baseline (years)	61.4 ± 13.5	65.5 ± 11.7	0.038	64.8 ± 13.6	67.1 ± 10.7	0.441
Sex			0.692			0.197
Female	62 (40.5)	21 (43.8)		23 (34.3)	14 (48.3)	
Male	91 (59.5)	27 (56.3)		44 (65.7)	15 (51.7)	
ECOG			0.360			0.915
0	50 (33.8)	22 (48.9)		27 (41.5)	13 (46.4)	
1	73 (49.3)	15 (33.3)		26 (40.0)	9 (32.1)	
2	18 (12.2)	5 (11.1)		8 (12.3)	4 (14.3)	
3	5 (3.4)	2 (4.4)		4 (6.2)	2 (7.1)	
4	2 (1.4)	1 (2.2)		0 (0.0)	0 (0.0)	
Cancer type			0.166			0.775
MM	78 (51.0)	18 (38.3)		36 (53.7)	13 (46.4)	
NSCLC	53 (34.6)	15 (31.9)		17 (25.4)	8 (28.6)	
RCC	7 (4.6)	7 (14.9)		3 (4.5)	1 (3.6)	
Other	15 (9.8)	7 (14.9)		11 (16.5)	6 (21.5)	
Stage UICC/AJCC			0.869			0.779
2	2 (1.4)	1 (2.2)		2 (3.2)	1 (3.4)	
3	12 (8.5)	3 (6.7)		10 (15.9)	3 (10.3)	
4	128 (90.1)	41 (91.1)		51 (81.0)	25 (86.2)	
Multiple Metastases (yes > 1 site)			0.103			0.144
Yes	117 (81.8)	28 (70.0)		49 (79.0)	16 (64.0)	
No	26 (18.2)	12 (30.0)		13 (21.0)	9 (36.0)	
PD-L1 status			0.835			0.987
Positive	32 (61.5)	16 (64.0)		13 (59.1)	10 (58.8)	
Negative	20 (38.5)	9 (36.0)		9 (40.9)	7 (41.2)	
Brain metastases			<0.001			0.610
Yes	52 (34.0)	2 (4.2)		4 (6.0)	1 (3.4)	
No	101 (66.0)	46 (95.8)		63 (94.0)	28 (96.6)	
Prior ipilimumab			0.033			0.458
Yes	34 (22.5)	4 (8.5)		8 (11.9)	2 (6.9)	
No	117 (77.5)	43 (91.5)		59 (88.1)	27 (93.1)	
Smoker			0.869			0.898
Yes	35 (25.0)	10 (26.3)		17 (25.4)	7 (24.1)	
No	105 (75.0)	28 (73.7)		50 (74.6)	22 (75.9)	
BMI			0.430			0.600
<18.5	4 (2.6)	2 (4.2)		1 (1.5)	1 (3.4)	
18.5–24.9	50 (32.7)	17 (35.4)		21 (31.3)	10 (34.5)	
25.0–29.9	56 (36.6)	19 (39.6)		28 (41.8)	14 (48.3)	
>30	30 (19.6)	4 (8.3)		17 (25.4)	4 (13.8)	
Pembrolizumab	75 (49.0)	18 (37.5)	0.163	36 (53.7)	11 (37.9)	0.155
Nivolumab	78 (51.0)	30 (62.5)		31 (46.3)	18 (62.1)	
Cycles	7.0 (3.0–16.0)	10.0 (1.0–19.0)	0.982	7.0 (3.0–16.0)	3.0 (1.0–17.0)	0.081
RT single dose	3.0 (2.0–9.0)	-	–	3.0 (2.0–3.0)	–	–
RT total dose	35.0 (20.0–45.0)	–	–	37.5 (28.5–50.0)	–	–
RT localization			–			–
Brain	51 (34.0)	0 (0.0)		14 (21.2)	0 (0.0)	
Lung	15 (10.0)	0 (0.0)		9 (13.6)	0 (0.0)	
Bone/soft tissue	49 (32.7)	0 (0.0)		22 (33.3)	0 (0.0)	
Lymph node	31 (20.7)	0 (0.0)		20 (30.3)	0 (0.0)	

Baseline characteristics for both treatment groups before and after propensity score matching (PSM). Continuous variables are reported as mean ± SD (normal distributed) or median (range) (not normal distributed) and categorical variables as *n* (%). Percentages are adjusted for missing values. For continuous variables, a Kruskal–Wallis test and for categorical variables, a Chi-squared test was carried out with α = 5%. *p* values are two sided. ECOG = Eastern Cooperative Oncology Group; MM = malignant melanoma; NSCLC = non-small cell lung cancer; RCC = renal cell carcinoma; PD-L1 = programmed death ligand 1; BMI = body mass index; ICI = immune checkpoint inhibitors; RT = radiation therapy.

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
