# Peer review of "Addition of Radiotherapy to Immunotherapy: Effects on Outcome of Different Subgroups Using a Propensity Score Matching"

_cancers, 2020, doi:10.3390/cancers12092429_

Round 1

Reviewer 1 Report

This paper presents the effects of RT in addition to PD-1 inhibition in a PS-matched analysis. Despite very contemporary and of significant interest,  I found the manuscript extremely difficult to follow and I believe it should be significantly simplified and totally re-organized. 

More specifically: 

  1. The population is very dishomogeneous and the presentation of the results is confusing with so many variables, making the reader dazed and with no clear take home message. I commend the authors for the attempt to systematize such a complex data-lake, but I think their efforts have not been paid as hoped and that a complete re-thinking of the paper is needed. Reading the manuscript is now difficult and the amount of information is not balanced in the different sections.
  2. The radiotherapy details are not well described: understanding fractionations and total doses is very difficult, limiting the impact of the observations and leaving important information out.
  3. The tables and graphs are too complex and definitely overcrowded: the reader will lose his bearings (e.g. figure 3 reports 10 curves while table 2 has 20 variables!)
  4. Discussion is the most interesting section but the overall take home message is penalized by the presentation of the methodology and results.  
  5. English should be improved and some typos and inconsistencies are present here and there.

Author Response

Dear reviewer,

thank you very much for your detailed review. 

We totally restructured and simplified the entire manuscript.

Please find our point-by-point response in the table below. 

We hope that we could meet all the requirements.

Yours sincerely,

Maike Trommer

Line number

Requested change

Exact change

Line 224 ff.(Safety);

Line 361 ff.;

Line 442 ff. (Table A2 and A3)

1. The population is very dishomogeneous and the presentation of the results is confusing with so many variables, making the reader dazed and with no clear take home message. I commend the authors for the attempt to systematize such a complex data-lake, but I think their efforts have not been paid as hoped and that a complete re-thinking of the paper is needed. Reading the manuscript is now difficult and the amount of information is not balanced in the different sections.

The inhomogeneous population is mainly due to the retrospective character of this study. The patients who received PD-1 inhibitors mostly underwent a variety of treatments and many of them were irradiated several times. To obtain reliable results, we considered multiple covariates to reduce this bias. This makes the analysis very complex. We have restructured the entire manuscript.

To keep the results clearer, we decided not to report about adverse events in detail and restructured this part in the results section. We restructured Tables 3 and 4 and attached those as appendix (Table A2 and A3). We have acknowledged the limitations of this study in the discussion.

1. Line 375 ff.;

2. Line 146 ff.;

3. Line 174 ff.;

4. Line 456 ff.

2. The radiotherapy details are not well described: understanding fractionations and total doses is very difficult, limiting the impact of the observations and leaving important information out.

We explained the different radiotherapy regimes used in the methods section and simplified the description of the radiotherapy details in the results section:

1. Methods:

"Possible fractionation schemes were the conventional fractionated radiation therapy (CFX), which is between 1.8 and 2 Gy single dose with 5 fractions per week, the hypofractionated radiation therapy (HFX) with higher irradiation doses between 2.5 and 4 Gy single dose and less fractions; the stereotactic body RT (SBRT) and the stereotactic radiosurgery (SRS) with ablative single doses between 9 and 20 Gy and one or very few fractions as very precise irradiation of small tumor volumes."

2. Results:

"Conventionally fractionated RT (CFX) was applied in 34.1 (44.8) % of the cases, n = 46 (26). Hypofractionated treatment (HFX) was overrepresented with 40.7 (41.4) %, n = 55 (24), SBRT and SRS (stereotactic body RT and stereotactic radiosurgery) was applied in 25.2 (13.8) % of the cases, n = 34 (8)."

3. Furthermore, we added the description of the outcome in the text and

4. the corresponding survival curves of CFX and SRS treated patients in the appendix (Fig. A2).

1. Line 99 ff.;

2. Line 124 ff.;

3. Line 442 ff. and 447 ff.;

4. Line 217 ff.

3. The tables and graphs are too complex and definitely overcrowded: the reader will lose his bearings (e.g. figure 3 reports 10 curves while table 2 has 20 variables!)

1. Table 1:

- We changed the 4 cutoffs for BMI to 2 cutoffs (<= 25 and >25).

- We summarized SCLC, Bladder CA, Esophageal CA, and Hodgkin's lymphoma as "other”.

- We deleted UICC/AJCC stage 1 as there were no patients with this stage.

- We deleted the Betablocker status, as this was only important for PSM in this study

2. Table 2:

- We summarized SCLC, Bladder CA, Esophageal CA, and Hodgkin's lymphoma as "other”.

- We deleted UICC/AJCC stage 1 as there were no patients with this stage.

- We deleted the Betablocker status

- We deleted RT type and timing from the table.

3. Tables 3 and 4 were restructured and are now Table A2 and Table A3 in the supplementary section.

4. Figure 3:

- We removed Fig. 3B (12 months OS) and show only the 50 months OS time plot (Fig. 3A)

- We removed Fig. 3C (Ipilimumab-naïve patients) and report this data only in the main text

- We removed Fig 3J (12 months OS regarding Malignant Melanoma) and display only the 50 months OS time plot for MM (Fig. 3I).

Line 234 ff.

4. Discussion is the most interesting section but the overall take home message is penalized by the presentation of the methodology and results.  

We amended this section.

5. English should be improved and some typos and inconsistencies are present here and there.

We revised that as requested and hope that we found all errors.

Reviewer 2 Report

Dear authors,

Thank you for providing a retrospective study of adding radiotherapy to PD-1 inhibitors in a variety of cancers. While the study premise is promising and provided information about the benefit of adding RT to ICB, it suffers from major drawbacks.

  1. The introduction is loosely written, with vague sentences, improper, insufficient, or lack of citations across the board. This a purely European study and the authors should clarify that at the beginning and in the materials and methods section. Uniklinik Koeln has not ben mentioned anywhere in the patient selection section.
  2. The results sections are almost impossible to follow with a lack of adequate referrals to the correct figure numbers especially in figures 2 and 3. Since the authors do not refer to the appropriate figure while describing the survival curve, the task falls on the reviewers to dig through each curve, detect and match the right curve to the text and then review. I stopped reviewing the manuscript after figure 2 since it was getting tedious. Please thoroughly refer to each curve while you describe the results.

Find my detailed review (up to the end of figure 2) attached.

Author Response

Dear reviewer,

thank you very much for the detailed review. We tried our utmost to address every single remark and hope that we meet all requirements. The entire manuscript has been restructured. 

Please find our point-by-point response in the table below and the point-by-point response to the detailed review attached.

Yours sincerely,

Maike Trommer

Line number

Requested change

Exact change

Line 39-40;

Line 87;

Line 96;

Line 368

1. The introduction is loosely written, with vague sentences, improper, insufficient, or lack of citations across the board. This a purely European study and the authors should clarify that at the beginning and in the materials and methods section. Uniklinik Koeln has not ben mentioned anywhere in the patient selection section.

We restructured the introduction, added citations where this was recommended and tried to eliminate any vaguenes. We restructured the results section, amended the methods section and adapted the discussion. We mentioned that we worked unicentric at the University Hospital of Cologne (Uniklinik Köln) and provide retrospective data of about 200 patients (see lines).

Line 93 ff. (Results);

Line 218 (Fig. 3);

Line 224 ff. (Safety);

Line 456 ff. (Tables A2 and A3)

2. The results sections are almost impossible to follow with a lack of adequate referrals to the correct figure numbers especially in figures 2 and 3. Since the authors do not refer to the appropriate figure while describing the survival curve, the task falls on the reviewers to dig through each curve, detect and match the right curve to the text and then review. I stopped reviewing the manuscript after figure 2 since it was getting tedious. Please thoroughly refer to each curve while you describe the results.

We restructured the results section and reduced Figure 3.

To keep the results clearer, we decided not to report about adverse events in detail and restructured this part in the results section. We restructured Tables 3 and 4 and attached those as supplementary Tables A2 and A3.

We referred to all figures and tables in the main text for description or discussion.

Reviewer 3 Report

The manuscript entitled "Effects on outcome and toxicity of radiotherapy in addition to PD-1 inhibition: a PS-matched analysis considering different subgroups" focused the attention on the role of RIT in different malignant entities.

Comments:

  • Introduction section is too long. Please could the Authors reduce this section?
  • The Authors adopted PD-L1 positive and negative. Could the Authors provide the three approved cut-off for PD-L1 expression (<1%, >= 1 <50, >= 50%).
  • Could the Authors better describe how PD-L1 IHC analysis has been performed (Antibody, platform)?
  • The Authors should control all the acronyms through the text and provide the extensive forms when they first appear.
  • Please use through the text "PD-L1" and "PD-1". Please control.

Author Response

Dear Reviewer,

thank you very much for your detailed review.

Please find our point-by-point response in the table below.

We hope that this meets all requirements.

Thank you very much and kind regards,

Maike Trommer

Line number

Requested change

Exact change

Line 55 ff.

  • Introduction section is too long. Please could the Authors reduce this section?

We reduced and restructured the introduction.

Line 171 ff.; 441 ff.

  • The Authors adopted PD-L1 positive and negative. Could the Authors provide the three approved cut-off for PD-L1 expression (<1%, >= 1 <50, >= 50%).

We provided the statistics for PD-L1 divided into the three approved cut-offs (<1%, >= 1 <50, >= 50%) and attached the PD-L1 expression curves as supplementary material (Fig. A1). In the unmatched patient cohort, 27 patients had a PD-L1 expression of <1%, 28 patients 1-50% and 13 patients >=50%. We had 133 missing values. In the patient group with a PD-L1 expression of 1-50% there is a trend towards a better OS when treated with radio-immunotherapy (p = 0.084).

  • Could the Authors better describe how PD-L1 IHC analysis has been performed (Antibody, platform)?

In NSCLC patients, the Tumor Proportion Score (TPS) was used with PD-L1 expressions of 0-100%. All tumor cells are counted that stain membrane-bound positive for PD-L1.

For all other cancer types, the Combined Positive Score (CPS) was used, which counts PD-L1 positive cells including lymphocytes and macrophages (0-100%).

  • The Authors should control all the acronyms through the text and provide the extensive forms when they first appear.

We revised that as requested.

  • Please use through the text "PD-L1" and "PD-1". Please control.

We revised that as requested.

Round 2

Reviewer 3 Report

I have no further comments.